# Logging Affects Genetic Diversity Parameters in an *Araucaria angustifolia* Population: An Endangered Species in Southern Brazil

**Rafael H. Roque** [1], **Alexandre M. Sebbenn** [2], **David H. Boshier** [3], **Afonso F. Filho** [4] **and Evandro V. Tambarussi** [1,5,*]

1 Programa de Pós-Graduação em Recursos Florestais, Escola Superior de Agricultura "Luiz de Queiroz", Universidade de São Paulo, Av. Pádua Dias, 11, Piracicaba 13418-900, SP, Brazil; rafahroque@usp.br
2 Instituto Florestal de São Paulo, São Paulo 01059-970, SP, Brazil; alexandresebbenn@yahoo.com.br
3 Department of Biology, University of Oxford, South Parks Road, Oxford OX1 3RB, UK
4 Universidade Estadual do Centro-Oeste (Unicentro), Irati 84505-677, PR, Brazil
5 Departamento de Produção Vegetal, Universidade Estadual Paulista (Unesp), Av. Universitária, 3780, Altos do Paraíso, Fazenda Experimental Lageado, Botucatu 18610-034, SP, Brazil
* Correspondence: evandro.tambarussi@unesp.br

**Abstract:** *Araucaria angustifolia* is an endangered species with more than 97% of its natural populations extinct. Logging of the species in the few remaining natural populations is highly restricted, though not readily accepted by farmers and logging companies. Consequently, political pressures have emerged for a return to logging of the species. Assessing the sustainability of such logging requires studies of a range of impacts on the remaining populations, including their genetic viability. We investigated the effect of selective logging on genetic diversity, intrapopulation spatial genetic structure (SGS), effective population size ($N_e$), and pollen and seed dispersal in three *A. angustifolia* permanent sample blocks established in a remnant of Araucaria Forest in Brazil. In these sample blocks, three logging intensities were applied (LI: 18.4, 31.4, and 32.3% of trees). Microsatellite analysis was performed for all adult and juvenile trees pre- and post-logging saplings. After selective logging, the greatest loss of alleles and the greatest decrease in $N_e$ were observed from the highest LI. Logging increased SGS, while the distance and patterns of pollen and seed dispersal were different for both pre- and post-logging scenarios, with pollen dispersed over greater distances than seed. Pollen dispersal distance post-logging and seed dispersal distance pre- and post-logging decreased with the increased distance between parents. After logging, $N_e$ reduced from 27.7 (LI = 31.4%) to 28.8 (LI = 18.4%) and 39.5% (LI = 32.3%), and some alleles were lost. Despite this, the loss of these alleles may be compensated for in subsequent generations, considering that logging resulted in changes such as an increase in the rate and distance of pollen immigration. Under the conditions evaluated in this study, selective logging of *A. angustifolia* is not adequate. To achieve truly sustainable forest logging, new rules that combine higher minimum DBH, lower logging intensity, and longer cutting cycles must be adopted. Furthermore, extensive genetic studies must be performed before logging any individual from a natural population.

**Keywords:** coniferous; forest logging; microsatellite markers; genetic conservation; Parana pine

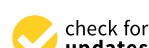



## 1. Introduction

For the selective logging of tree species, whether in natural or planted populations, the current local legislation must be considered, since in some countries, such as Brazil, native species can only be logged from natural forest based on a sustainable management plan, which stipulates parameters such as minimum cutting diameter (MCD), logging intensity (LI) above the MCD, and length of cutting cycles (CC) [1–6]. However, such plans usually neglect genetic aspects, although there are a number of possible impacts both short and long

term. Selective logging may have two main genetic effects on natural populations: (1) a genetic bottleneck due to the reduction in the total number of reproductive individuals of the same generation; (2) genetic drift due to the increase in the distance among reproductive trees remaining after logging, changing the mating and gene dispersal patterns, and consequently the gene frequencies of succeeding generations [2]. These effects can have negative genetic impacts on harvested species such as the loss of alleles and decrease in heterozygosity [7–9] and the increase in self-fertilization and fixation index [10–13]. By contrast, positive effects have also been observed, such as decreases in biparental inbreeding, correlated mating [14], and intrapopulation spatial genetic structure (SGS) [15,16].

Analysis of genetic diversity allows inferences about selective logging impacts, in terms of allelic loss, decreases in heterozygosity, and effective population size ($N_e$), as well as changes in mating and gene dispersal patterns and SGS [7,17,18]. These population genetic parameters relate to a species' population demography (number and density of reproductive trees, regeneration rate, recruitment, survival) which together with commercially important growth traits (diameter, height, stem form, and volume) provide the necessary information to determine limits, such as MCD, LI, and CC, for sustainable logging.

In this context, *Araucaria angustifolia* (Bertol.) Ktze. (Araucariaceae) is a species of great ecological, economic, and social importance within its natural occurrence, with a history of intense exploitation due to its wood quality and growth potential [19]. Endemic to Argentina, Brazil, and Paraguay (latitude 19°15′ S–31°30′ S, longitude 41°30′ W–54°30′ W, altitude 211–2400 masl) [19,20], *A. angustifolia* is a wind-pollinated conifer with a dioecious reproductive system, although in rare cases, monoecious individuals have also been observed [21,22]. In natural populations, individuals reach reproductive age after approximately 20 years [23], with a period between fertilization and cone maturation ranging from 20 to 24 months [24] or even more than 30 months [25]. Seeds are initially dispersed by barochory, with secondary dispersal by animals such as *Dasyprocta* spp. (rodents) and *Cyanocorax caeruleus* (bird) [26,27], generally between 60–80 m from mother trees [19]. Population density is variable among sites, ranging from 5 to 44.2 trees ha$^{-1}$ [19,28,29], following an inverted J-shape distribution for diameter at breast height (DBH), with a high frequency of trees in small diameter classes, decreasing in frequency with an increase in diameter [30–32].

*Araucaria angustifolia* is long-lived, living more than 300 years, reaching more than 250 cm DBH and up to 50 m in height [19]. The species is a dominant tree of the Mixed Ombrophilous Forest (Araucaria Forest) in southern Brazil, considered the most endangered Brazilian biome [33]. Until the 19th century, Araucaria Forest occupied 20 million hectares in southern Brazil [34]. Through the 20th century, until the 1970s, Araucaria Forest was considered the most important ecosystem in southern Brazil, underpinning a large part of the forest timber economy, with *A. angustifolia* itself one of the most important timber species, due to its high growth rate and timber value [35,36]. Due to extensive exploitation, mainly from the 1950s to 1970s, Araucaria Forest has suffered a strong retraction in its distribution with remnants currently occupying only 3 to 13% of the original area [27]. Thus, much of the species' natural populations and their genes have already become extinct. *A. angustifolia* is thus designated as critically endangered globally by the International Union for the Conservation of Nature's Red Book [37] and as "vulnerable", according to the Brazilian Ministry of the Environment official list of endangered flora. Currently, logging within Araucaria Forest is highly restricted for most species, especially those listed as endangered, such as *A. angustifolia*, although permitted under exceptional circumstances, such as for environmental licensing [4]. However, these restrictions are not always well accepted by small producers and logging companies, and political pressures have recently emerged to allow selective logging through the application of reduced impact logging (RIL), which consists of a range of techniques to mitigate the impacts of selective logging [4].

For *A. angustifolia*, there are some studies of selective logging [38–40] and of genetic diversity [30,32,41–44], but few studies of logging impacts on genetic diversity [4,8,45].

Research is necessary due to the large genetic differences observed among populations, such that different populations of the same species may respond differently to selective logging [29,44,46–49]. Furthermore, it is important to study and develop logging rules that, in addition to conserving natural resources, also generate economic returns, especially for the small producers typical of much of its distribution in Brazil [50]. To provide a sound basis to maintain or change current rules for selective logging of *A. angustifolia* requires studies of the impacts, including genetic diversity, on logged populations [4].

Our study aimed to evaluate the effects of selective logging on genetic diversity, SGS, pollen and seed dispersal, and $N_e$ of a natural *A. angustifolia* population and to propose strategies for the species' sustainable logging. We address the following questions: (i) How does selective logging affect genetic diversity parameters and SGS of a natural population? (ii) Is there a difference in the genetic diversity of adults, juveniles, and saplings? (iii) How does selective logging affect gene flow in the species? (iv) Are the current logging intensity regulations adequate for sustainable management of the species?

## 2. Material and Methods

### 2.1. Site and Sampling Design

The study was carried out in an experimental area of 20 ha, which is part of a fragment of the Legal Reserve of Araucaria Forest, covering 130 ha in Fernandes Pinheiro, Paraná state, Brazil (Figure 1), and which was studied by the Universidade Estadual do Centro-Oeste (Unicentro), in the "Imbituvão project". One of the project's aims was to develop a model of sustainable forest management for Araucaria Forest remnants, which could be replicated in southern Brazil. The experimental area was inventoried in 2014 for trees with a minimum DBH of 30 cm, resulting in 1000 identified individuals (50 ha$^{-1}$). The area was subdivided into 20 blocks of 1.0 ha, with the following traits considered for logging: DBH > 30 cm, phytosanitary quality, stem quality, crown diameter, and volume, estimated according to [51]. Trees of high wood volume and wood and phytosanitary quality were prioritized for logging, resulting in a reduced DBH across the remaining individuals compared to the original population (Supplementary Materials, Table S1). The distance between trees was also considered to avoid opening very large clearings, harming the development of other individuals, and to conserve the forest's species diversity. Based on the density of trees, three of the 20 blocks were chosen to study the selective logging: B13 with highest density (132 ind ha$^{-1}$ > 30 cm, 90 ind ha$^{-1}$ > 40 cm DBH), B15 with medium density (85 ind ha$^{-1}$ > 30 cm, 47 ind ha$^{-1}$ > 40 cm DBH), and B20 with the lowest density (63 ind ha$^{-1}$ > 30 cm, 44 ind ha$^{-1}$ > 40 cm DBH), totaling 3 ha (Figure 1). A total of 48 trees > 40 cm DBH were logged from the three blocks between May and July 2016, with a similar logging intensity used in the contiguous blocks, B13 (32.4%) and B15 (31.4%), whereas in block B20, a lower logging intensity was used (18.4%). LI was 70% higher in B13 (mean DBH of logged trees = 53.6 cm, ranging from 40 to 67.8 cm) and in B15 (DBH = 48.6 cm, 39.5–62.4 cm) than in B20 (DBH = 53.2 cm, 44.6–64.9 cm), but resulted in a smaller decrease in DBH post-logging (B13 = 4.3%, B15 = 5.5% and B20 = 2.8%) than pre-logging.

For the genetic analyses, a different number of individuals were sampled in each of the three blocks. From 2017 (157 adults > 30 cm of DBH) to 2019 (78 juveniles and 115 saplings), we sampled 350 individuals from the total *A. angustifolia* population, of which 45 were logged, while the others were from the remnant population. Three of the 48 logged individuals were not available for sampling. We collected cambium from the logged trees, samples of leaves and cambium from the remnant adult trees, and leaves from juveniles and saplings. Saplings were those with a height of up to 2 m and a diameter of 2 to 3 cm. Juvenile individuals were those with a height of 6 to 12 m and a diameter ranging from 15 to 40 cm. Additionally, and more importantly, juveniles are distinguished by a crown shape resembling a "cup", while adults are over 20 m tall, exhibit reproductive structures, and have a "candelabra" crown shape. Regardless of diameter, individuals are considered adults when they show reproductive structures and can have different sexes. Individual saplings were sampled after mid-2019, and the smallest ones (height from 18

to 73 cm, diameter from 2.2 to 6.9 mm) prioritized to ensure that they truly represented a post-logging reproductive event. Juvenile individuals ranged from 3.5 to 22.6 m in height, and 2.4 to 37.8 cm for DBH. We prioritized sampling saplings close to logged trees, usually in groups of six to eight, where seedling density was visibly higher from the resultant post-logging gaps. Other saplings, as well as juveniles, were also randomly sampled within the blocks. Geographical coordinates were recorded for all 350 individuals (Figure 1) using a global positioning system (GPS Garmin Etrex, available from garmin.com, accessed on 5 May 2020), while mapping used ArcGIS Pro software (available from esri.com/arcgis, accessed on 8 August 2021).

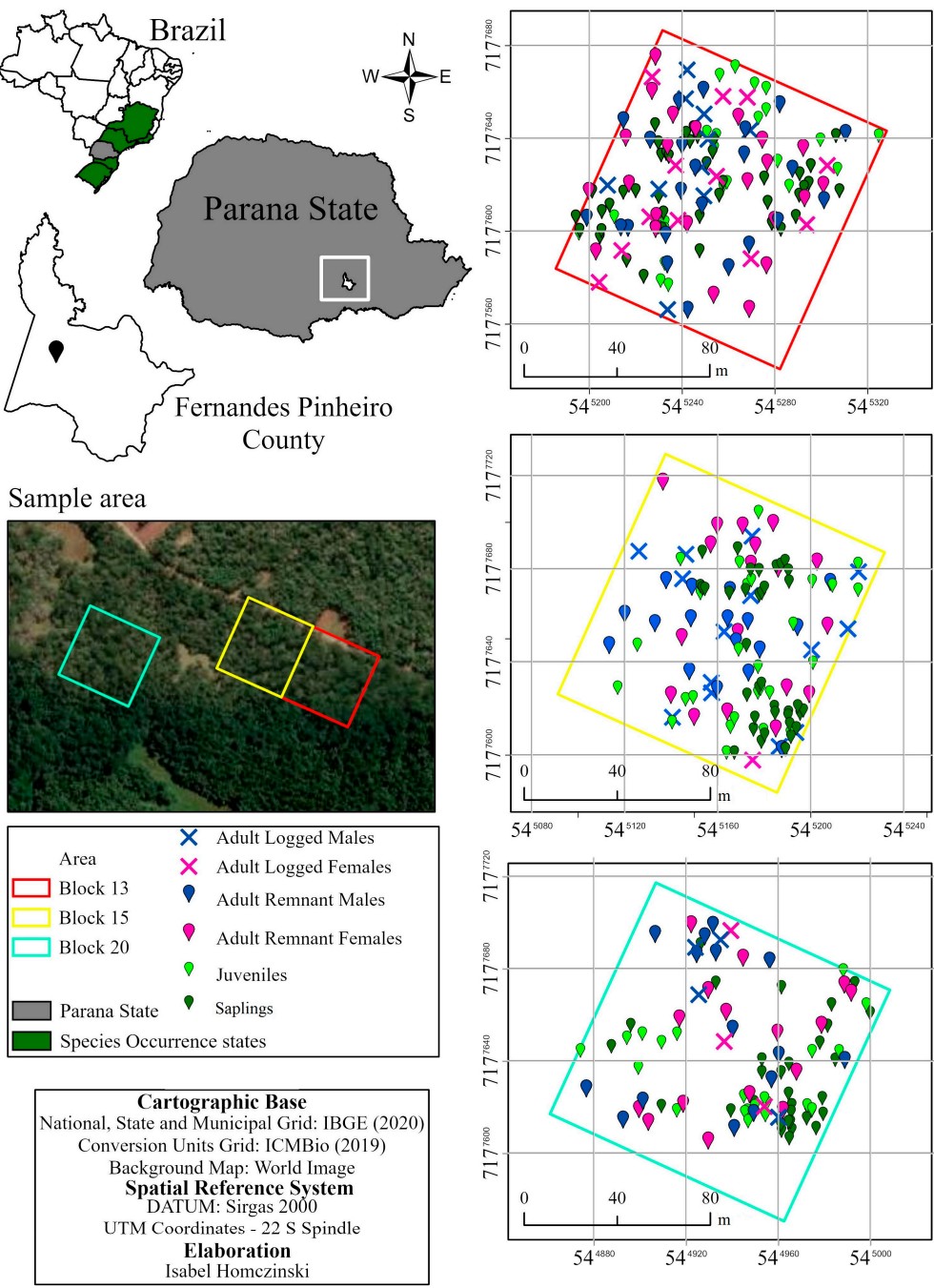

**Figure 1.** *Araucaria angustifolia* population study located in Fernandes Pinheiro county, Paraná State, Brazil. Blocks 13, 15, and 20 represent different logging intensities in the experiment, totaling a 3 ha sample area. In the figure, only adults, juveniles, and saplings that were genotyped are represented.

## 2.2. DNA Extraction and Microsatellite Genotyping

After collection, samples were stored in thermal boxes containing ice and transported to the Heréditas laboratory (Technology in DNA Analysis, Brasília, Distrito Federal, Brazil) for DNA extraction and genotyping. DNA extraction followed the protocol of [52], with extraction of total genomic DNA from about 150 mg of fresh tissue macerated in liquid nitrogen [53]. DNA was quantified, determined, and run on an agarose gel 1% stained with ethidium bromide and visualized under ultraviolet light. Thirteen microsatellite markers developed and described by [54] (Aang1, Aang12, Aang14, Aang15, Aang27, Aang28, Aang37, Aang43), [55] (Ag20, Ag56, Ag62) and Grattapaglia (unpublished: Aa1774, Aa53325) were tested for the population.

## 2.3. Analysis of Genetic Diversity

Linkage disequilibrium between pairs of loci was tested across all samples to verify the allelic association of different loci, with statistical significance assessed using Monte Carlo permutations and a Bonferroni correction ($\alpha = 0.05$). Genetic diversity was estimated for adults, juveniles, and adults + juveniles (A + J) pre-logging, and adults, saplings, and adults + juveniles + saplings (A + J + S) post-logging by blocks (B13, B15, and B20), for total number of alleles (K), allelic richness (R), observed ($H_o$), and expected ($H_e$) heterozygosity under Hardy–Weinberg equilibrium. To investigate if there was evidence for inbreeding in the samples, the mean fixation index (F) was estimated and statistical significance tested using a permutation of alleles between individuals, associated with a Bonferroni correction for multiple comparisons (95%, $\alpha = 0.05$). All estimates were obtained using the FSTAT 2.9.4 software [56]. The presence of null alleles was estimated for loci showing a fixation index significantly higher than zero in adults, juveniles, and saplings under a population inbreeding model (PIM), using INEST 2.2 software [57]. Allele frequencies were used to estimate the number of private alleles in pre-logging adults and juveniles, post-logging adults, juveniles and saplings, and the number of alleles lost due to logging, comparing pre- and post-logging adults.

## 2.4. Spatial Genetic Structure and Effective Population Size

Spatial genetic structure (SGS) was estimated in blocks B13, B15, and B20 for adults pre- and post-logging and total samples of pre- (adults + juveniles) and post-logging (adults + juveniles + saplings), using the coefficient of coancestry ($\theta_{ij}$) [58] within different distance classes and the SPAGEDI 1.5a software [59]. To compare SGS between blocks for adults (pre- and post-logging), adults + juveniles (pre-logging), and adults + juveniles + saplings (post-logging), due to differences in the number of individuals between the pre- and post-logging stages (within and between blocks), distance classes were defined as similarly as possible across the blocks to maintain at least 100 pairs of individuals within classes (see SPAGEDI 1.5a, User's Manual, 4.4. Information displayed during computations). In only two cases was the 100-pair rule violated (0–25 m class for post-logging adults at B15 (98 pairs) and B20 (75 pairs; Supplementary Materials Table S2). The paired mean $\theta_{ij}$ was plotted against the geographic distance between pairs of individuals and the statistical significance of mean values for each distance class used to compare confidence limits to 95% of the mean, by permuting 1000 individuals between distance classes, for the null hypothesis of no SGS ($\theta_{ij} = 0$).

As we were only able to determine the sex of adult trees, estimates of group coancestry ($\Theta$) and effective population size ($N_e$) were only made for the adults in each block. Group coancestry [60] for adults was estimated as the mean coefficient of coancestry among all pairs of adults ($\Theta$), considering the mean individual fixation index ($F_i$, assuming negatives values as zero), based on the Loiselle model [58] and using SPAGEDI 1.5a software [59]:

$$\Theta = \frac{\sum_{x=1}^{n_f} \sum_{y\neq 1}^{n_f} \theta_f}{4n_f^2} + \frac{\sum_{x=1}^{n_m} \sum_{y\neq 1}^{n_m} \theta_m}{4n_m^2} + \frac{\sum_{x=1}^{n_f} \sum_{y=1}^{n_m} \theta_{fm}}{2n_f n_m}$$

where $n_f$ and $n_m$ are the number of adult females and males, respectively; and $\theta_f$, $\theta_m$, and $\theta_{fm}$ are the coancestry coefficient between females, males, and females and males, respectively. $N_e$ was estimated by, $N_e = 0.5/\Theta$ [60].

The hypothesis of reduction in the effective population size, and consequently a reduction in genetic diversity due to forest management, was tested based on the relationship between allele frequencies and heterozygosity, using Bottleneck software, version 1.2 [61]. For this analysis, we used the Wilcoxon test with $\alpha = 0.05$ and the stepwise mutation model (SMM).

### 2.5. Parentage Analyses

Estimates of pollen and seed immigration and dispersal distance were made for each block by categorical parentage analysis (paternity and maternity, respectively), implemented in CERVUS 3.0.7 software [62]. Cryptic gene flow was calculated by $C_{GF} = 1 - P_{ex}^n$, where $P_{ex}$ is the combined probability of the nonexclusion probability of the first parent (probability of nonexclusion of a candidate female or male parent without the genotype of the opposite sex), calculated using CERVUS, and n is the number of putative female and male parents sampled in the population [63]. As the species is wind pollinated and seed dispersal occurs initially by gravity, animals can act as secondary seed dispersers [27] and the two blocks are located close to each other (mean, minimum, and maximum distance for females and males between B13 and 15 and B20 of 280, 124, and 425 m, respectively, Supplementary Materials, Table S3), parentage analyses were carried out separately for each block. However, all sampled pre-logging adult female and male trees from both blocks were used as putative pollen (fathers) or ovule (mothers) donors of juveniles in the two blocks, while only the female and male trees of both blocks remaining post-logging were used as putative mother and father of saplings. To account for unsampled adults, we set the proportion of genotyped pollen donors within each block as 70% within CERVUS. The Δ statistic was used to assign the most likely pair of parents, female or male parents, based on the allele reference frequencies of all genotyped individuals, whose significance was determined from the CERVUS paternity simulation. The critical Δ value was determined for an 80% confidence level, using 10,000 repetitions, a genotyping error ratio of 0.01, 50% as the proportion of pollen and ovule donors sampled within each block, and a minimum number of nine loci for parentage assignment. When the analysis was unable to designate one of the potential parents within the population, the juvenile or seedling was considered to represent pollen or seed immigration. Based on the georeferenced coordinates (x, longitude; y, latitude) of all adults, juveniles, and saplings, we calculated the minimum, maximum, mean, and median pollen and seed dispersal distances, based on the Euclidean distance between two points. Pollen dispersal was estimated for juveniles and saplings assigned simultaneously for both female and male parents, by the distance between both.

## 3. Results

### 3.1. Genetic Diversity

No significant linkage genotypic disequilibrium was detected after a Bonferroni correction ($p < 0.0003$) between pairs of loci (Supplementary Materials, Table S4), indicating that this set of loci is suitable for population genetic studies due to the random association of alleles between loci, which is a pre-requisite to avoid bias in estimates of genetic parameters. The estimate of null allele frequency was only significantly higher than zero in adults for loci Aang12 (Supplementary Materials, Table S5). Across the total sample of adults, juveniles, and saplings in all blocks, a total (K) of 143 alleles were found. Seven, nine and five alleles were private in adults pre-logging in B13, B15, and B20, respectively, whereas four alleles were private in juveniles pre-logging in B15, and two, three, and five alleles were private in saplings post-logging in B13, B15, and B20, respectively (Table 1). Logging resulted in the loss of ten, seven, and four alleles in B13 (7.2%), B15 (5.4%) and B20 (3.1%), respectively (all with frequency < 0.05), whereas only four, two, and one of these alleles were not found post-logging (adults + juveniles + saplings) in B13, B15 and

B20, respectively (Table 2). No significant differences were observed for all blocks between adults, juveniles, A + J, saplings, and A + J + S pre- and post-logging for allelic richness (R), observed ($H_o$), and expected ($H_e$) heterozygosity, and fixation index (F) (Table 1). The F values were significantly lower or not different from zero in all samples and blocks, indicating absence of inbreeding in the samples pre- and post-logging.

**Table 1.** Genetic diversity pre- (Pre) and post-logging (Post) for adults, juveniles, and saplings for blocks B13, B15, and B20 in the *Araucaria angustifolia* population.

| | n | K | $P_k$ | R (95% CI) | $H_o$ (95% CI) | $H_e$ (95% CI) | F (95% CI) |
|---|---|---|---|---|---|---|---|
| B13 (LI = 32.3%) | | | | | | | |
| Pre: adults (A) | 68 | 138 | 7 | 9.1 (6.4–11.9) | 0.821 (0.774–0.868) | 0.770 (0.686–0.854) | −0.066 * (−0.165–0.021) |
| Post: adults (A) | 46 | 128 | 0 | 9.0 (6.4–11.7) | 0.812 (0.762–0.861) | 0.769 (0.685–0.852) | −0.056 (−0.147–0.035) |
| Pre: juveniles (J) | 25 | 105 | 0 | 8.0 (5.7–10.3) | 0.830 (0.723–0.937) | 0.750 (0.650–0.850) | −0.107 (−0.167–0.047) |
| Pre: A + J | 93 | 136 | - | 9.0 (6.3–11.6) | 0.824 (0.764–0.885) | 0.769 (0.682–0.856) | −0.072 * (−0.151–0.007) |
| Post: saplings (S) | 43 | 131 | 2 | 9.0 (6.4–11.7) | 0.785 (0.671–0.898) | 0.753 (0.656–0.850) | −0.042 (−0.093–0.009) |
| Post: A + J + S | 114 | 136 | - | 8.8 (6.3–11.4) | 0.806 (0.724–0.887) | 0.762 (0.671–0.853) | −0.057 * (−0.102–0.012) |
| B15 (LI = 31.4%) | | | | | | | |
| Pre: adults (A) | 51 | 130 | 9 | 9.2 (6.3–12.1) | 0.813 (0.745–0.881) | 0.769 (0.683–0.856) | −0.057 (−0.122–0.008) |
| Post: adults (A) | 35 | 123 | 0 | 9.2 (6.3–12.0) | 0.831 (0.773–0.890) | 0.772 (0.690–0.855) | −0.076 (−0.144–0.008) |
| Pre: juveniles (J) | 30 | 117 | 4 | 8.9 (6.3–11.5) | 0.783 (0.690–0.877) | 0.751 (0.661–0.841) | −0.043 (−0.079–0.007) |
| Pre: A + J | 81 | 137 | - | 9.3 (6.4–12.1) | 0.802 (0.726–0.877) | 0.765 (0.677–0.854) | −0.048 (−0.097–0.001) |
| Post: saplings (S) | 40 | 118 | 3 | 8.6 (6.0–11.4) | 0.799 (0.695–0.904) | 0.744 (0.652–0.837) | −0.074 (−0.129–0.019) |
| Post: A + J + S | 105 | 138 | - | 9.1 (6.3–11.9) | 0.805 (0.721–0.890) | 0.759 (0.670–0.848) | −0.061 * (−0.101–0.021) |
| B20 (LI = 18.4%) | | | | | | | |
| Pre: adults (A) | 38 | 129 | 5 | 9.2 (6.6–11.7) | 0.761 (0.717–0.789) | 0.780 (0.682–0.878) | 0.025 (−0.051–0.101) |
| Post: adults (A) | 31 | 125 | 0 | 9.1 (6.5–11.7) | 0.779 (0.686–0.872) | 0.778 (0.674–0.882) | −0.001 (−0.080–0.078) |
| Pre: juveniles (J) | 23 | 108 | 0 | 8.3 (6.0–10.6) | 0.796 (0.685–0.906) | 0.744 (0.640–0.849) | −0.069 (−0.130—0.006) |
| Pre: A + J | 61 | 129 | - | 8.8 (6.4–11.3) | 0.774 (0.684–0.864) | 0.766 (0.666–0.866) | −0.011 (−0.069–0.047) |
| Post: saplings (S) | 32 | 120 | 5 | 8.5 (5.9–11.1) | 0.772 (0.642–0.902) | 0.749 (0.634–0.863) | −0.031 (−0.098–0.036) |
| Post: A + J + S | 86 | 133 | - | 8.7 (6.2–11.2) | 0.781 (0.674–0.888) | 0.759 (0.653–0.865) | −0.029 (−0.078–0.020) |

n is the sample size; LI is logging intensity; K is the total number of alleles; $P_k$ is the number of private alleles; R is the allelic richness for 24, 28, and 23 multilocus genotypes in B13, B15, and B20, respectively; $H_o$ and $H_e$ are the observed and heterozygosity, respectively; F is the fixation index; * $p < 0.05$.

**Table 2.** Number and frequency of private and loss alleles due to the logging in adults (A) pre- and post-logging, juveniles (J), and saplings (S) in the *Araucaria angustifolia* blocks.

| | Block 13 | Block 15 | Block 20 |
|---|---|---|---|
| Logging intensity (%) | 32.3 | 31.4 | 18.4 |
| Total number of alleles in adults pre-logging | 138 | 130 | 129 |
| Total number of alleles in adults post-logging | 128 | 123 | 125 |
| Number of loss allele in adults post-logging (%) | 10 (7.2) | 7 (5.4) | 4 (3.1) |
| Maximum allele frequency | 0.022 | 0.029 | 0.028 |
| Number of loss alleles in adults present in juveniles | 3 | 4 | 2 |
| Number of loss alleles in adults present in saplings | 5 | 4 | 2 |
| Number of loss alleles in adults present in A + J + S | 6 | 5 | 3 |
| Number of alleles not found within blocks | 4 | 2 | 1 |
| Maximum allele frequency | 0.015 | 0.029 | 0.027 |

## 3.2. Spatial Genetic Structure and $N_e$

Before logging, generally no significant spatial genetic structure was observed in any of the blocks, except for B20 where A + J showed significant SGS between 0–15 m and 25–50 m. Logging resulted in significant SGS in all blocks across most assessed groupings (Figure 2). The increase in SGS post-logging was reflected in group coancestry (Θ) for adults which increased from pre- to post-logging in all blocks, resulting in lower $N_e$ post- (22.4–28.2) than pre-logging (37–39), especially in B13, where the highest LI was applied

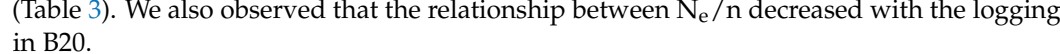

(Table 3). We also observed that the relationship between $N_e/n$ decreased with the logging in B20.

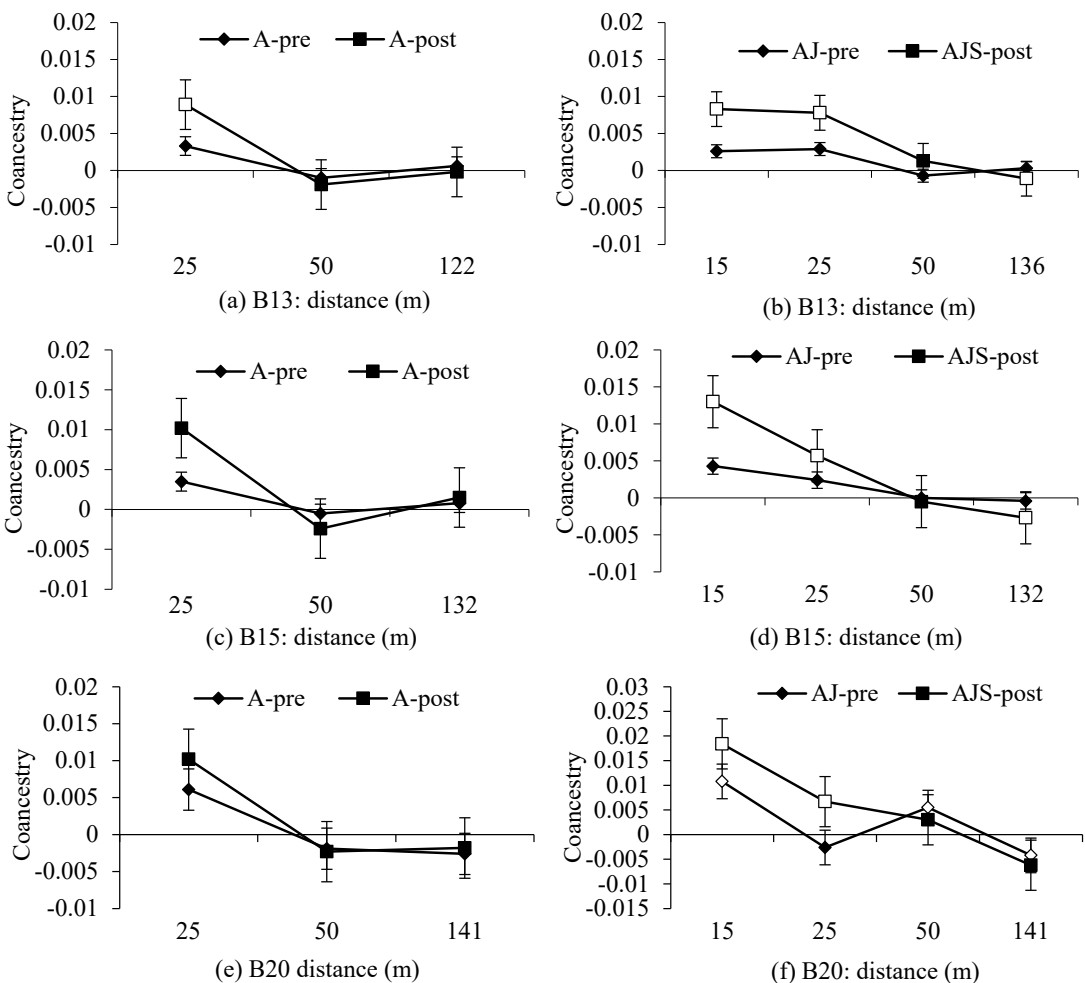

**Figure 2.** Spatial genetic structure in *Araucaria angustifolia* for adults pre- (A-pre) and post-logging (A-post) (**a,c,e**), adults and juveniles pre-logging (AJ-pre), and adults, juveniles, and saplings post-logging (AJS-post) (**b,d,f**) in blocks B13, B15, and B20; open symbols indicate mean value of the coancestry coefficient significantly different from zero.

**Table 3.** Sample size, group coancestry ($\Theta$), and effective population size ($N_e$) for adults in pre- and post-logging in blocks B13, B15, and B20.

| | $n_M{:}n_F$ | n | M:F | $\theta_{MM}$ | $\theta_{FF}$ | $\theta_{FM}$ | $\Theta$ | $N_e$ | $N_e/n$ | Decrease in $N_e$ (%) |
|---|---|---|---|---|---|---|---|---|---|---|
| **B13** | | | | | | | | | | |
| Pre | 33:35 | 68 | 1:0.94 | 0.0038 | 0.0097 | −0.0016 | 0.0135 | 37 | 0.54 | 39.5 |
| Post | 23:23 | 46 | 1:1 | 0.0087 | 0.0114 | 0.0022 | 0.0224 | 22.4 | 0.49 | |
| **B15** | | | | | | | | | | |
| Pre | 31:20 | 51 | 1.55:1 | 0.0064 | 0.0113 | −0.0049 | 0.0128 | 39 | 0.76 | 27.7 |
| Post | 17:18 | 35 | 1:0.94 | 0.0113 | 0.0119 | −0.0085 | 0.0177 | 28.2 | 0.81 | |
| **B20** | | | | | | | | | | |
| Pre | 19:19 | 38 | 1:1 | 0.0066 | 0.0066 | 0.00004 | 0.0132 | 37.9 | 0.99 | 28.8 |
| Post | 15:16 | 31 | 1:1.07 | 0.0093 | 0.0092 | −0.0023 | 0.0185 | 27 | 0.87 | |

$n_M$, $n_F$, and n are the number of males, females, and total sample size, respectively; $\theta_{MM}$, $\theta_{FF}$, and $\theta_{FM}$ are the mean pairwise coancestry between males, females, and males and females, respectively; decrease (%) in $N_e$, $= 100[1 - (N_{e(Post)}/N_{e(Pre)})]$.

The genetic bottleneck analysis showed that for two of the blocks (B13, B20) the hypothesis of excess heterozygotes (presence of a bottleneck) was not significant in the pre-logging scenario (B13 $p = 0.080$, B20 $p = 0.188$), while it was significant in the post-logging scenario (B13 $p = 0.004$, B20 $p = 0.019$). In B15, for both pre- ($p = 0.040$) and post-logging ($p = 0.024$), the hypothesis of excess heterozygotes was significant.

Parentage analysis was carried out for pre-logging juveniles and post-logging saplings (Table 4 and Figure 3). The $P_{ex}$ (0.999) value indicates that these loci are suitable for use in *A. angustifolia* parentage studies. Pollen immigration and mean dispersal distance were higher than seed immigration and dispersal distance both pre- and post-logging, with most seeds moving short distances (within 25 m), while pollen moves much further (>300 m), particularly post-logging (Figure 3). Pollen immigration ($m_p$) from outside blocks was generally higher post-logging (74.4%–96.9%) than pre- (53.3%–78.3%), whereas seed immigration ($m_s$) decreased from pre- (13%–30%) to post-logging (5%–18.6%). The distance and pattern of pollen dispersal were different from seed dispersal in both pre- and post-logging scenarios, with pollen generally dispersed longer distances than seeds (Table 4 and Figure 3). The exception was post-logging in B13, where seed dispersal reached a greater distance (313 m) than pollen dispersal (243 m). Mean pollen dispersal distances in B13 and B20 were similar in pre- (84 and 187 m) and post-logging (68 and 178 m), whereas in B15 they increased from pre- (85 m) to post-logging (112 m). Mean seed dispersal distances decreased in all block from pre- (55–68 m) to post-logging (23–32 m). Mean post-logging pollen dispersal distances were higher than median values in B13 and B15, as well as pre-logging in B15, while mean seed dispersal distances (pre- and post-logging) were also greater than median values in all blocks. These results indicate a pattern of isolation by distance, with pollen dispersed principally to near neighbors and seed dispersed close to the mother tree.

**Table 4.** Realized immigration of pollen ($m_p$) and seed ($m_s$) within block and dispersal distance pre- (juveniles) and post-logging (saplings) in *Araucaria angustifolia* blocks.

| | | Pollen Dispersal (m) | | | | Seed Dispersal (m) | | |
|---|---|---|---|---|---|---|---|---|
| | $m_p$ | Mean | Median | Min/max | $m_s$ | Mean | Median | Min/max |
| Block13 | | | | | | | | |
| Pre: juveniles | 76 | $84 \pm 29$ | 89 | 9/162 | 28 | $55 \pm 13$ | 50 | 6/112 |
| Post: saplings | 74.4 | $68 \pm 31$ | 44 | 2/243 | 18.6 | $32 \pm 18$ | 18 | 2/313 |
| Block15 | | | | | | | | |
| Pre: juveniles | 53.3 | $85 \pm 31$ | 59 | 9/233 | 30 | $56 \pm 38$ | 45 | 4/155 |
| Post: saplings | 85 | $112 \pm 38$ | 97 | 21/270 | 5 | $25 \pm 6$ | 18 | 2/97 |
| Block20 | | | | | | | | |
| Pre: juveniles | 78.3 | $187 \pm 51$ | 219 | 41/368 | 13 | $68 \pm 43$ | 29 | 2/366 |
| Post: saplings | 96.9 | $178 \pm 56$ | 187 | 4/349 | 9.4 | $23 \pm 17$ | 12 | 3/268 |

Min/max is the minimum and maximum dispersal distance; $\pm$ is the 95% standard error, 1.96 SE.

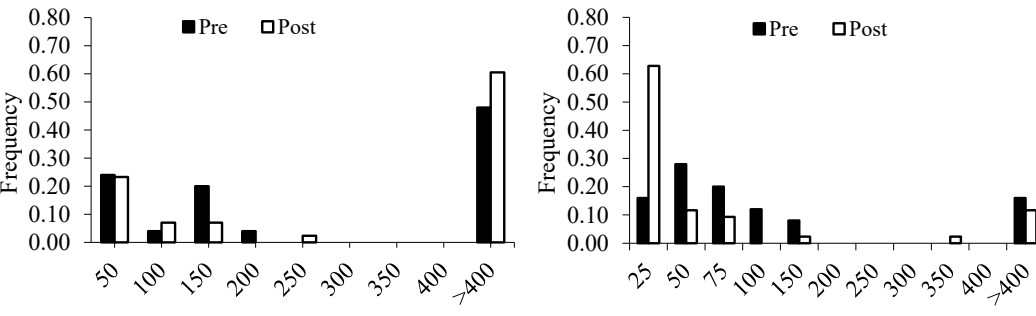

(a) B13: pollen dispersal distance (m)

(b) B13: seed dispersal distance (m)

**Figure 3.** *Cont.*

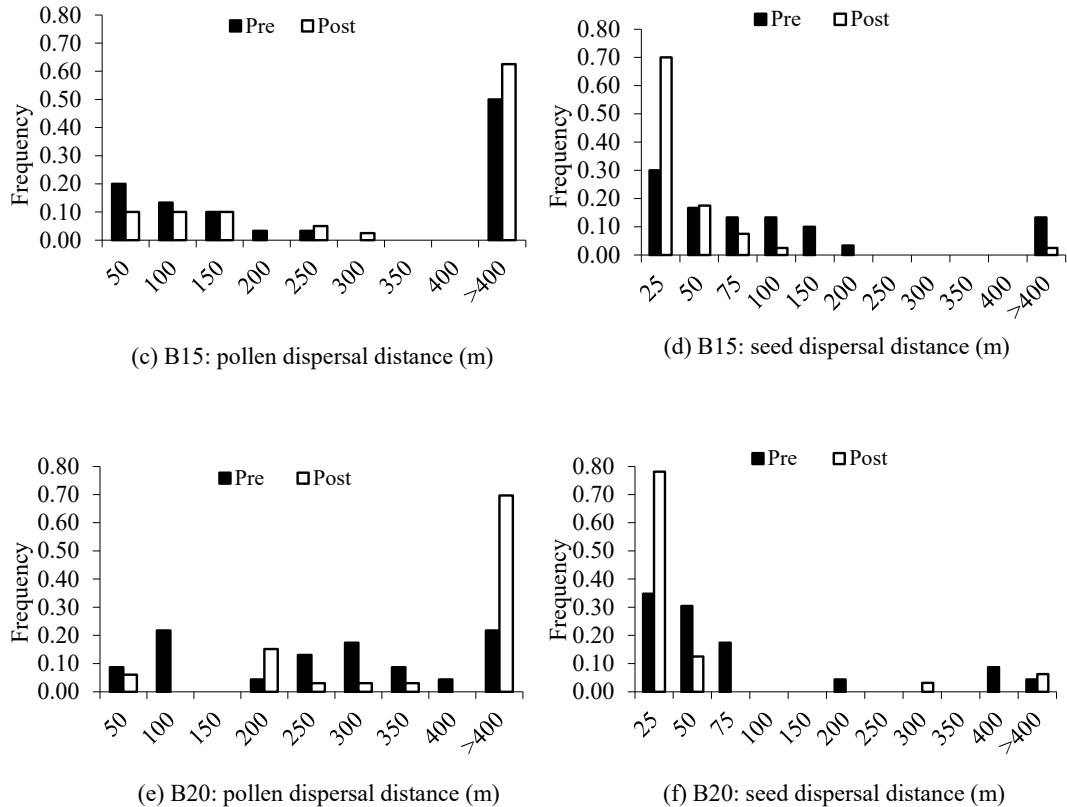

(c) B15: pollen dispersal distance (m)

(d) B15: seed dispersal distance (m)

(e) B20: pollen dispersal distance (m)

(f) B20: seed dispersal distance (m)

**Figure 3.** Frequency distribution for observed distance dispersal events of pollen (**a**,**c**,**e**) and seed (**b**,**d**,**f**) dispersal, determined by parentage analysis of Araucaria angustifolia sampled in pre- and post-logged blocks B13, B15, and B20.

## 4. Discussion

In this study, the impacts of selective logging on genetic diversity, spatial genetic structure (SGS), effective population size ($N_e$), and pollen and seed flow in three experimental blocks established within a natural population of *A. angustifolia* were investigated, comparing pre- and post-logging scenarios, using 13 SSR loci. The main observed impacts of logging on genetic diversity were a loss of low frequency alleles, a decrease in $N_e$, seed immigration and dispersal distance, and increases in pollen immigration and SGS, with generally the most pronounced impacts being observed at the highest logging intensities.

### 4.1. Genetic Diversity and Effective Population Size

The loss of alleles and decrease in $N_e$ in adults due to logging is a clear result of a genetic bottleneck, where some alleles occurring at low frequency (<0.05) in pre-logging adults are lost and $N_e$ decreases due to the reduction in population size. Logging resulted in the largest loss of alleles for blocks B13 and B15, which were submitted to the highest LI. However, the majority of the alleles lost from the logged adults were still present in the juveniles and saplings within those same blocks, showing that juveniles established pre-logging and pollen and seed immigration post-logging from nearby blocks and unlogged forest minimized the effect of allele loss from logged adults. Another notable result is the fact that pre-logging juveniles already have fewer alleles than adults, which may suggest that genetic drift in subsequent generations may be independent of logging. No differences were observed between individual groups for allelic richness (R) and observed ($H_o$) and expected ($H_e$) heterozygosity in both blocks, but special attention should be given to the allelic richness results obtained by rarefaction. Allelic richness is associated with the total number of alleles (K) and may indicate that the latter is affected by sample size [64] such that the results may change with a larger sample of individuals.

Similar to our results, other studies have observed the loss of low frequency alleles due to selective logging in tree populations of several species in Brazil [2–5,12,15,16,65–70] and around the world [17,71–83], and no impacts on $H_o$ and $H_e$, which are measures less sensitive to logging [2,75,83–88]. Clearly, logging has a greater impact on allelic diversity than on heterozygosity. Although the alleles lost in adults were generally present in the juvenile and seedling cohorts, there is a long way to go before these cohorts reach adulthood. Some individuals may not survive to adulthood, due to stochastic events or deterministic processes.

The decrease in $N_e$ in adults post-logging can be explained by the decrease in population size and increase in group coancestry ($\Theta$), as many logged trees were not genetically related, whereas many unlogged trees are genetically related. Post-logging, within each block the remaining trees have higher $\Theta$ than pre-logging, which means a greater probability that individuals within these blocks share identical alleles by descent (IBD), resulting in a lower $N_e$. These results may be unfavorable for maintaining the population's genetic diversity, as they imply a smaller number of individuals contributing to the total genetic variability, given the reduction in $N_e$, in addition to the possibility of higher mating rates between related individuals due to an increase in coancestry.

### 4.2. Gene Flow and Spatial Genetic Structure

The results indicate a pattern of isolation by distance, with pollen dispersed principally to near neighbors and seed dispersed close to the mother tree. Logging had little impact on the distance between female and male trees (Supplementary Materials, Table S3). The decrease in population size due to logging resulted in an increase in pollen immigration, whereas mean pollen dispersal distance was similar within blocks. The higher post-logging pollen immigration in the three blocks and the immigration and pollen dispersal distances in B20 being greater than in B13 and B15 could be due to logging decreasing the number of reproductive trees within blocks. B20 has fewer reproductive trees than the other blocks (B13,15), both pre- and post-logging, and showed higher pollen immigration and dispersal distances both pre- and post-logging. Post-logging, higher pollen immigration into B15 and B20 and dispersal distances in B15 may also be explained by the fact that saplings are subjected to natural selection (e.g., against related mating) and stochastic processes (e.g., random mortality, disease, and predation) for a shorter time than juveniles.

Our estimates for pollen immigration were higher and mean dispersal distances lower than reported for juveniles and saplings of the same species in a 5.4 ha forest fragment (immigration = 3%–6.5%; dispersal distance = 70–75 m) [30], in a 7.4 ha plot within continuous forest (immigration = 5.9%, dispersal distance = 134 m) [32], and a 16.2 ha plot within a grassland area (immigration = 23.8%; dispersal distance = 415 m) [89]. Furthermore, long-distance pollen dispersal in *A. angustifolia* has been reported between populations within forest fragments and isolated trees (up to 2000 m), although a high frequency of mating occurs at short distances, ranging from 25 to 298 m [30,32,41,89–91]. Pollen dispersal in *A. angustifolia* occurs by wind and high-density forest may act as a barrier for pollen dispersal, decreasing pollen dispersal distances [91]. Reducing population size by selective logging reduces plant density, which may reduce barriers to pollen dispersal, increasing gene flow. This may explain the higher post-logging pollen immigration observed by [32] in a continuous forest.

By contrast, logging decreased both seed immigration and mean dispersal distance, although mean distance was not strongly different between blocks pre- and post-logging, generally favoring seedling establishment near mother trees. Maximum seed dispersal distance did increase in B13 post-logging, though saplings generally established about a third of the distance of pre-logging. This probably occurred due to *A. angustifolia* being a pioneer species that regenerates within forest gaps, such that the gaps created within the blocks by logging favor seedling regeneration closer to mother trees [38].

Seed immigration pre- and post-logging was lower than reported for the same species in a 5.4 ha forest fragment (immigration = 10%) [30], and not strongly different to

that in a 7.4 ha continuous forest (5.9%) [32] and a 16.2 ha plot within a grassland (immigration = 11%) [89]. Mean seed dispersal distance was lower than reported for juveniles and saplings in other studies, ranging from 92 to 229 m [30,32,89]. The higher seed dispersal distances in the other studies can be explained by their larger sample areas, allowing detection of long-distance dispersal events. Even the mean seed dispersal distances were different from those in our study, though the pattern of dispersal was the same, with a greater frequency of juveniles and saplings established near mother trees [30,32].

The greater observed pollen than seed immigration and dispersal distance, both pre- and post-logging, is explained by the fact that *A. angustifolia* pollen, although unsaccate and generally larger than other coniferous wind-pollinated tree species [92], can be dispersed over longer distances than seeds, which are primarily dispersed by barochory and secondarily by animals [27,30,32].

Spatial genetic structure is a common phenomenon in *A. angustifolia* populations, mainly originating from the short seed dispersal distance, as reported in other studies of this species [32,42,44,89,90,93,94]. The increase in SGS post-logging was due to seedling establishment near the mother tree, resulting in greater spatial seedling aggregation in regeneration, favored by the clearings originating from logging and reflecting the species' heliophyte nature. *Araucaria angustifolia* regeneration is greater in open and secondary forest than mature forest [30,32,89]. However, studies suggest that the SGS in juveniles and saplings decreases through to the adult stage, probably as a result of stochastic effects as random mortality, predation, diseases, or deterministic causes, such as selection against inbred individuals or inbreeding depression [30,32].

*4.3. Considerations for Management*

The results of this study can inform in situ and ex situ conservation strategies, particularly related to seed collection for genetic improvement, management, and environmental reforestation. The statistically significant SGS up to 25 m, detected for post-logging adults, indicates that, theoretically, seed collection in this and similar populations should be carried out from trees separated by more than this distance. However, the present study was carried out in 1.0 ha blocks, limiting the possibility of observing the occurrence of SGS above 141 m. Other studies of *A. angustifolia* have considered larger sample areas, ranging from 5.4 to 14 ha, detecting statistically significant SGS from 25 to 201 m [30,32,42,89,90,93,94]. Therefore, it is suggested that *A. angustifolia* seed collections be carried out at distances of at least 100 m between seed trees. This sampling strategy can ensure high genetic diversity in the seed, as there will be a lower probability of genetically related adults at this distance, in addition to sampling a greater number of different pollen clouds.

Logging intensity was more than 70% higher in B13 (32.9% of trees logged, mean DBH of logged trees = 53.6 cm) and B15 (31.4%, DBH = 48.6) than in B20 (18.4%, DBH = 53.2 cm). The genetic and demographic ECOGENE model [95] was previously used to investigate the long-term effects of selective logging on basal area (BA) and genetic diversity of *A. angustifolia* populations to determine selective logging scenarios that ensure sustainability of timber yield [4]. The study found that for a MCD of 50 cm (similar to the mean used here) and LI of 20, 40, and 90%, BA recovered to the original value after 54, 74, and 84 years, respectively. Genetic distance estimates showed that the greatest changes in allele frequencies occurred with the use of 50 cm MCD. With pre-logging BA not recovering after 40 years for an MDC of 50 cm, the simulations showed that current Brazilian legislation (MCD = 50 cm, LI = 90%, CC = 40 years) does not lead to sustainable timber production of *A. angustifolia*. Based on such results, and the mean logging DBH of about 50 cm (minimum of 40 cm and maximum of 67.8 cm) used here and LI of 18.4% (B20), 31.4% (B15), and 31.9% (B13), we can expect basal area recovery in B20 in about 50 years (18.4% × 54 years/20%) and 59 years in B13 and B15 (31.9% × 74 year/40%), respectively.

## 5. Conclusions

Logging decreases allelic diversity, effective population size, seed immigration and dispersal distance, but generally increases SGS and pollen immigration. Saplings contribute to genetic diversity; however, it is not possible to guarantee that they will survive and reach reproductive age. The post-logging increase in coancestry in the first distance class indicates that selective logging is increasing seedling establishment near the mother tree, whereas trees that were logged favored seed dispersal over long distances.

We can also consider that if the population continues to be exploited following the same criteria, it may continue to be exposed to processes such as loss of genetic diversity through genetic bottlenecks and changes in pollen and seed dispersal patterns, affecting subsequent generations. So, the most viable option would be to prioritize the selection of trees for logging close to each other (closer than 25 m), which are probably more related, to avoid losses of genetic diversity. This will facilitate the flow of pollen, favoring crossing between unrelated individuals and contributing to the population's genetic diversity.

It is important to highlight the levels of genetic diversity observed both in juveniles, which were already part of the pre-logging population, and in the saplings, which appeared after logging and were probably favored due to the opening of the canopy. This genetic diversity can compensate for allelic losses generated by selective logging; however, there is no guarantee that the individual saplings will reach reproductive age, when they would contribute to the $N_e$. Thus, it is very important that the development of juveniles and saplings is monitored, as it is fundamental for the maintenance of allelic diversity within the population and from the perspective of ensuring that there is adequate post-logging regeneration.

Under the conditions evaluated and considering the results from our study, current selective logging rules are not adequate for *A. angustifolia*. We detected that selective logging resulted in $N_e$ reduction even at the lowest logging intensity, as well as the loss of some alleles. Furthermore, our results reinforce the view that sustainable timber production of *A. angustifolia* cannot be achieved with current cutting cycles of 40 years or less, as permitted in both Brazil and Argentina. It is evident that the legislation in both these countries is ineffective in terms of the conservation of *A. angustifolia*, mainly because it does not differentiate between species—one size does not fit all! Nor does the legislation address the full range of relevant issues; instead, it emphasizes production and omits aspects such as genetic conservation and environmental education [4,8].

However, given the economic importance of this species, we must consider the likelihood of noncompliance with the ban on logging the species, and consequently the possibility of illegal exploitation of its populations, factors that can harm the species' conservation. This study shows that the logging of native tree species populations should not be based only on dendrometric criteria or on current Brazilian legislation, but should apply the principles outlined here to also ensure genetic viability. Furthermore, it is evident that detailed genetic and ecological studies must be performed before logging any individual from natural populations of threatened tree species.

**Supplementary Materials:** The following supporting information can be downloaded at: https://www.mdpi.com/article/10.3390/f14051046/s1, Table S1: Number of *Araucaria angustifolia* adults, females, males, juveniles, and seedlings sampled (n), and logging intensity (LI) in blocks in scenarios of pre-logging (Pre), logged, and post-logging (Post) in three sample blocks (B13, B15, B20) and total sample of a remnant of Araucaria Forest located in Fernandes Pinheiro County, Parana state, Brazil; Table S2: Sample size (*n*) and number of pairs ($N_{pairs}$) of adults (A), adults + juveniles (A + J) and adults + juveniles + seedlings (A + J + S) pre- and post-logging by distance classes used in the analysis of the spatial genetic structure in blocks B13, B15 and B20; Table S3: Mean, minimum, and maximum distance between females (FF), males (MM), and females and males (FM) within and among blocks B13, B15 and B20 in pre- (Pre) and post-logging (Post) sceneries; Table S4: Results of *p*-values for linkage genotypic disequilibrium for all *Araucaria angustifolia* samples (adults + juveniles + seedlings); Table S5: Genetic diversity per locus and mean loci, estimate of frequency of null alleles ($Freq_{(null)}$) and the fixation index uncorrected (*F*) and corrected for null alleles ($F_{null}$)

for 13 microsatellite loci analyzed in pre- and post-logging scenarios in an *Araucaria angustifolia* population located in Fernandes Pinheiro County, Parana state, Brazil.

**Author Contributions:** R.H.R. was responsible for data collection, literature review, data analysis, interpretation of results, and text writing. A.M.S. was responsible for the research design, methodology, data analysis, supervision, and writing of the text. D.H.B. was responsible for designing the research and correcting and supervising the text writing. A.F.F. was responsible for project administration and review of the text. E.V.T. was responsible for project administration, conceptualization, visualization, supervision, and review and editing of the text. All authors have read and agreed to the published version of the manuscript.

**Funding:** This research was financed by resources from the extension project "Strategies for forest management in small rural properties in the Center-South of Paraná" (SETI/UGF), from the Universidade Estadual do Centro-Oeste. The authors were supported by grants from Conselho Nacional de Desenvolvimento Científico e Tecnológico (CNPq) for doctoral studies (grant number 14020/2020-2) and research productivity fellowships (grant numbers 304899/2019-4 and 304650/2020-0). Evandro V. Tambarussi was also supported by a post-doctoral scholarship (grant number 200727/2020-6) from CNPq.

**Data Availability Statement:** Genotype (https://doi.org/10.6084/m9.figshare.19643274) and phenotype data (https://doi.org/10.6084/m9.figshare.19643283) have been submitted to the Figshare Digital Repository.

**Acknowledgments:** Rafael H. Roque received a scholarship from the Conselho Nacional de Desenvolvimento Científico e Tecnológico (CNPq). Evandro V. Tambarussi and Alexandre M. Sebbenn were supported by research productivity fellowships granted by CNPq. We would like to thank the Universidade Estadual do Centro-Oeste for funding the research project: "Strategies for forest management in small rural properties in the Center-South of Paraná" (grant from Recursos do Estado do Paraná (SETI/UGF)). We thank Dario Grattapaglia and Heréditas for genotyping the plant material. We thank our colleagues Isabel Homczinski, Rafael Ferreira, Alexandre Garret, Marcio Teixeira, and Tiago Grespan for help with collecting data in the field. We thank everyone who supported this research.

**Conflicts of Interest:** The authors declare no conflict of interest.

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
