# Peer review of "Logging Affects Genetic Diversity Parameters in an Araucaria angustifolia Population: An Endangered Species in Southern Brazil"

_forests, doi:10.3390/f14051046_

Round 1

Reviewer 1 Report

The paper focuses on the effects of logging intensities on genetic diversity, effective population size, SGS and gene flow in Araucaria angustifolia in Brazil by genotyping 350 individuals at 13 nuclear microsatellite loci. They conclude that under the conditions evaluated in this study, selective logging of A. angustifolia was not adequate due to a decrease in allelic diversity, ??, seed immigration and dispersal distance and an increase in SGS. 

The manuscript covers an interesting and worthwhile topic in a threatened species. Analytical methods are generally sound, but the methodological design is not adequate to fulfill the question about the effect of logging intensity. The sampled sites are 3, which differ in the initial density of individuals, present two logging intensities (because the B13 and B15 blocks have the same intensity) and do not present repetitions. So, it is difficult to determine if the differences found are due to logging intensities or other factors. Block B20 has the lowest logging intensity but presents also the lowest density of individuals both before and after logging. For that reason, some parts of interpretation of the results and discussion seem not reasonable.

Considering the great effort made in the study, it would be necessary to lower the level of the conclusions in order to consider its publication. I suggest re-framing the study to fit what the experimental design can deliver, analyzing the genetic diversity, effective population size, SGS and gene flow in logged areas but not accurately analyzing the effect of logging intensities.

 I also suggest to improve grammar and language. Some sentences are too long or confuse and this makes it difficult to understand the content.

 Different sections should be modify based on previous recommendations. However, I have some points that deserve the authors' attention (more comments are in the attached pdf):

            Title should be modified.

            In the methodology describe more clearly that the individuals analyzed are only a sample of those present in each plot. Define the classification of adults, juveniles and seedlings. Specify the ranges of diameters and heights for adults and juveniles

•           In results section include confidence intervals for Ne and mp or ms that allow determining if the differences found between the treatments (increases or decreases between pre and post) are significant. On the other hand, it is difficult for me to understand if there is a real loss of alleles due to the cutting, since the allelic richness when the rarefaction is carried out does not present differences.

 •           The discussion is too long; it should be shortened. Some of the sentences are speculative, they should be more consistent with what was studied. Some sentences lack fluency. I also suggest joining sections 4.3 and 4.4, management considerations should be made based on the results obtained in the study, e.g. distances between individuals due to logging should not be greater than the dispersal distances or, for ex situ conservation strategies if seeds are collected from logged areas they should be collected from individuals more than 30 m apart to avoid collecting seeds from genetically related individuals.

I encourage the authors to language editing, an intensive review of grammar issues throughout the manuscript should benefit it. 

Author Response

Comments and Suggestions for Authors #1

The paper focuses on the effects of logging intensities on genetic diversity, effective population size, SGS and gene flow in Araucaria angustifolia in Brazil by genotyping 350 individuals at 13 nuclear microsatellite loci. They conclude that under the conditions evaluated in this study, selective logging of A. angustifolia was not adequate due to a decrease in allelic diversity, ??, seed immigration and dispersal distance and an increase in SGS. 

The manuscript covers an interesting and worthwhile topic in a threatened species. Analytical methods are generally sound, but the methodological design is not adequate to fulfill the question about the effect of logging intensity. The sampled sites are 3, which differ in the initial density of individuals, present two logging intensities (because the B13 and B15 blocks have the same intensity) and do not present repetitions. So, it is difficult to determine if the differences found are due to logging intensities or other factors. Block B20 has the lowest logging intensity but presents also the lowest density of individuals both before and after logging. For that reason, some parts of interpretation of the results and discussion seem not reasonable.

Considering the great effort made in the study, it would be necessary to lower the level of the conclusions in order to consider its publication. I suggest re-framing the study to fit what the experimental design can deliver, analyzing the genetic diversity, effective population size, SGS and gene flow in logged areas but not accurately analyzing the effect of logging intensities.

 I also suggest to improve grammar and language. Some sentences are too long or confuse and this makes it difficult to understand the content.

 Different sections should be modify based on previous recommendations. However, I have some points that deserve the authors' attention (more comments are in the attached pdf):

  • Title should be modified.

Answer: New suggested title: Logging affects genetic diversity parameters in an Araucaria angustifolia, population: an endangered species in southern Brazil

  • In the methodology describe more clearly that the individuals analyzed are only a sample of those present in each plot. Define the classification of adults, juveniles and seedlings. Specify the ranges of diameters and heights for adults and juveniles

Answer: Regarding sampling in each block, we include the following: “For the genetic analyses, a different number of individuals were sampled in each of the three blocks”.

Regarding the classification of individuals, we include the following: “Seedlings  were those with a height of up to 2 m and a diameter of 2 to 3 cm. Juvenile individuals were those with a height of 6 to 12 m and a diameter ranging from 15 to 40 cm. Additionally, and more importantly, juveniles are distinguished by a crown shape resembling a “cup”, while, adults are over 20 m tall, with reproductive structures, and have a "candelabra" crown shape. Regardless of diameter, individuals are considered adults when they show reproductive structures and can have different sexes”.

  • In results section include confidence intervals for Ne and mp or ms that allow determining if the differences found between the treatments (increases or decreases between pre and post) are significant. On the other hand, it is difficult for me to understand if there is a real loss of alleles due to the cutting, since the allelic richness when the rarefaction is carried out does not present differences.

Answer: Regarding allelic richness, we include the following discussion: “Special attention should be given to the allele richness results obtained by rarefaction. Allelic richness is associated with the total number of alleles (K) and may indicate that the latter is affected by sample size (Kalinowski 2004) such that the results may change with  a larger sample of individuals.”

  • The discussion is too long; it should be shortened. Some of the sentences are speculative, they should be more consistent with what was studied. Some sentences lack fluency. I also suggest joining sections 4.3 and 4.4, management considerations should be made based on the results obtained in the study, e.g., distances between individuals due to logging should not be greater than the dispersal distances or, for ex situ conservation strategies if seeds are collected from logged areas, they should be collected from individuals more than 30 m apart to avoid collecting seeds from genetically related individuals.

Answer: We chose to keep sections 4.3 and 4.4 separate, as section 4.3 contains recommendations for management but also discusses our results with respect to the literature. We reserve item 4.4 for our direct conclusions. However, we have tried to reduce the length of the discussion.

Reviewer 2 Report

The manuscript titled Logging intensity effects on genetic diversity, effective population size, spatial genetic structure and gene flow in Araucaria angustifolia (Araucariaceae) has provided us many interesting information through microsatellite technology.

 The following opinions and suggestions are for reference.

 1.     Line 201.  I think the Formula is not correct, please check it!

 2.     Line 240.  “Seven” should be “7”?

 3.     In my opinion, density of adult trees has strong impact on pollen dispersal, so I hope that the density data related can be shown in the manuscript,

 4.     If possible, the bottleneck analysis is hoped to be conducted in the manuscript. It can provide direct evidence to logging effect.

Author Response

Comments and Suggestions for Authors #2

The manuscript titled Logging intensity effects on genetic diversity, effective population size, spatial genetic structure and gene flow in Araucaria angustifolia (Araucariaceae) has provided us many interesting information through microsatellite technology.

 The following opinions and suggestions are for reference.

  1. Line 201.  I think the Formula is not correct, please check it!

Answer: This has been checked and corrected.

  1. Line 240.  “Seven” should be “7”?

Answer: This was written in full because it was starting a sentence, as per general practice.

  1. In my opinion, density of adult trees has strong impact on pollen dispersal, so I hope that the density data related can be shown in the manuscript.

Answer: Tree density data are presented in section 2.1.

  1. If possible, the bottleneck analysis is hoped to be conducted in the manuscript. It can provide direct evidence to logging effect.

Answer: This has been implemented. The analysis showed a significant result for genetic bottleneck after logging.

Round 2

Reviewer 1 Report

Comments and suggestions for authors:

There is an extra comma in the title.

Despite the suggestions in the previous revision to change the focus of the study from the effect of logging intensities to the effect of logging, the summary was not modified.

I suggest replacing seedlings with saplings, since seedlings typically refer to plants in the early stages of development, shortly after germination, while saplings are more mature and according to the sizes referred to in this study.

Some suggestions and questions made in the manuscript in the previous revision were not taken into account, answered, or justified, and therefore my doubts about certain statements remain unanswered. The main one is this: "In results section include confidence intervals for Ne and mp or ms that allow determining if the differences found between the treatments (increases or decreases between pre and post) are significant". Therefore, some statistical analysis or confidence intervals must be included to determine if the higher or lower values of Ne, mp or ms are significantly different. If it is not included, I cannot affirm that logging effectively generate a reduction in Ne, they are only descriptive appreciations and not with a statistical basis. Even more I believe that there is an error in the ratio Ne/n in B15, the post logging value is higher than that shown and higher than the pre logging value.

Line 286. I do not believe that the decrease in the mean dispersal distance of pollen for B20 between 187 m +-51 and 178 m+- 56 (table 4) is large enough to report that it decreased after logging, moreover considering the standard error.

English was not reviewed and the same generalizations and speculations marked in the previous version continue in this version (line 328), without explaining or clarifying if it was previously studied or if it can be referenced.

Lines 329-330. Remove this sentence, it is not informative.

Lines 333-335. Clarify this sentence.

Line 344. Mp was not higher in the three blocks after logging.

Lines 344-346. Immigration and pollen dispersal distances were higher in B20 than B13 and B15 even before logging because it had lower density both before and after logging, it is not due to a logging effect

Line 349. Post logging, pollen dispersal distances were not higher in B13.

Line 434. I am not sure about this statement. I think this could be confirmed if seedlings in areas without logging had been studied. I consider this increase is the result of the development stage of seedlings and not of an effect of logging.

 Finally, I think there are many interesting analyzes but the discussion goes beyond what the results allow. This should be limited to what was actually found.

I still consider that English should be revised.

Author Response

Dear Mr. Vatavu V Vlad Viorel Assistant Editor Based on the final reviewer comments We are submitting our revised version of the manuscript "Logging affects genetic diversity parameters in an Araucaria angustifolia, population: an endangered species in southern Brazil". We have described in detail our responses to the final reviewer`s comments and hope that the manuscript now meets your expectations, and it can be accepted.
We thank the reviewers for their constructive criticism and suggestions.
Sincerely,
Evandro V. Tambarussi
